# The sources and diurnal variations of submicron aerosols in a coastal-rural environment near Houston, US

Jing Li<sup>1</sup>, Jiaoshi Zhang<sup>1</sup>, Xianda Gong<sup>1</sup>, Steven Spielman<sup>2</sup>, Chongai Kuang<sup>3</sup>, Ashish Singh<sup>3</sup>, Maria A. Zawadowicz<sup>3</sup>, Lu Xu<sup>1,\*</sup>, Jian Wang<sup>1,\*</sup>

<sup>1</sup>Center for Aerosol Science and Engineering, Department of Energy, Environmental and Chemical Engineering, Washington University in St. Louis, St. Louis, Missouri, USA;

Correspondence to: Jian Wang (jian@wustl.edu) or Lu Xu (xu1@wustl.edu)

10 Abstract. Aerosol properties were characterized at a rural site southwest of Houston from May to September 2022 during the intensive operation periods (IOP) of the Tracking Aerosol Convection Interactions ExpeRiment (TRACER). Backward trajectory analysis reveals three major air mass types, including marine air mass from the Gulf, urban air mass influenced by urban emissions, and regional air mass. Marine aerosols typically show a bimodal size distribution and have the lowest particle number and mass concentrations of PM<sub>1</sub> (particulate matter with an aerodynamic diameter of less than 1µm), while the aerosols from air masses 15 strongly influenced by urban emissions exhibit the highest concentrations. Organic aerosol (OA) accounts for more than 50% of PM<sub>1</sub> for urban and regional air masses, whereas sulfate is comparable to OA in marine air masses. Positive Matrix Factorization (PMF) analysis of aerosol mass spectra identifies 6 OA factors, including hydrocarbon-like OA (HOA), OA from the oxidation of monoterpenes (MT-SOA), OA from the reactive uptake of isoprene epoxydiols by acidic sulfate particles (isoprene-SOA), oxygenated OA arising from shipping emissions (shipping-OOA), and two oxygenated OA factors with high O:C ratios (OOA1 20 and OOA2). OOA2 has the highest O:C ratio and exhibits elevated mass concentration in the afternoon. Similar diurnal variation of highly oxidized OA factors was commonly observed in the Houston area during previous studies and attributed to the SOA formation by photochemistry and mixing from aloft. Here, using air mass backward trajectories and 1-D box model, we show the diurnal trend of OOA2 mass concentration is instead driven by changes in air mass arriving at the rural site. The air mass changes are likely caused by the shift between land breezes and sea/bay breezes. Within the same air mass type (e.g., either urban or marine 25 air mass), OOA2 mass concentration is largely independent of wind direction and shows essentially no diurnal variation, suggesting OOA2 is related to aged OA with minimal influence by local emissions. This study helps identify the major sources of OA in the Houston region and highlights the impacts of both atmospheric chemistry and meteorology on aerosol properties in the coastalrural environment.

<sup>&</sup>lt;sup>2</sup>Aerosol Dynamics Inc., Berkeley, California, USA.

<sup>&</sup>lt;sup>3</sup>Environmental and Climate Sciences Department, Brookhaven National Laboratory, Upton, NY, USA

## 1 Introduction

Aerosol particles can affect Earth's radiation budget by absorbing and scattering radiation in the atmosphere (direct effect) and affecting cloud albedo and lifetime via serving as cloud condensation nuclei and ice nuclei (indirect effects) (Albrecht, 1989; Charlson et al., 1992; Twomey, 1977). Aerosol can also influence convective clouds and precipitation (Andreae et al. 2004; Fan et al. 2007a; Fan et al. 2007b; Heever et al. 2006; Rosenfeld et al. 2008). The effects of aerosols on clouds are among the most significant uncertainties in the simulation of climate change since pre-industrial time (IPCC, 2023). In addition, aerosols are air pollutants and pose severe health risks when inhaled, contributing to respiratory and cardiovascular diseases (Lelieveld et al., 2015; Pope and Dockery, 2006). Quantifying these effects of aerosols on climate and human health requires the knowledge of the physical and chemical properties of aerosols, which are diverse spatiotemporally. Understanding the sources, precursors, and evolution of aerosols is essential to quantifying the properties and effects of aerosols, and their temporal and spatial variations.

Houston is the fourth largest city in the United States and has active energy and chemical sectors. The Port of Houston is one of the busiest seaports in the United States, with significant emissions from ships and heavy-duty diesel engines. The areas around Houston have abundant vegetation, including large forested areas to the north of the city. Isolated convective systems are common in the Houston region. The circulation of land and sea/bay breezes also plays an important role in shaping the atmospheric environment in the Houston area (Caicedo et al., 2019; Li et al., 2020; Wang et al., 2022). As a result, Houston experiences a spectrum of aerosol conditions, from those strongly influenced by urban, forested, and/or industrial emissions to significantly lower aerosol concentrations southwest of the city. Previous studies showed that OA and sulfate are the most abundant aerosol components in the Houston region during the summertime (Al-Naiema et al., 2018; Cleveland et al., 2012; Dai et al., 2019; Dunker et al., 2019; Leong et al., 2017; Schulze et al., 2018; Wallace et al., 2018; Yoon et al., 2020, 2021), similar to aerosol compositions in other coastal cities (Hersey et al., 2011; Kompalli et al., 2020; Lei et al., 2020; Liu et al., 2019; Wong et al., 2022). PMF analysis has been widely used to investigate OA sources in Houston, identifying a diverse range of contributing factors (Al-Naiema et al., 2018; Bean et al., 2016; Brown et al., 2013; Cleveland et al., 2012; Dai et al., 2019; Schulze et al., 2018; Wallace et al., 2018; Yoon et al., 2020, 2021; Zhou et al., 2023). Primary organic aerosol (POA) factors are predominately associated with anthropogenic emissions. Major sources of POA include fossil fuel combustion from vehicular traffic (Al-Naiema et al., 2018; Cleveland et al., 2012; Wallace et al., 2018) and shipping activities (Schulze et al., 2018). Additionally, other sources such as cooking emissions and biomass burning emissions (Dai et al., 2019; Wallace et al., 2018) were identified. Less oxidized secondary organic aerosol (SOA) factors have been linked to the oxidation product of biogenic emissions (Brown et al., 2013), based on their characteristic mass spectral signatures. Highly oxidized SOA factors were consistently observed in Houston, often accounting for a substantial fraction of the OA mass concentrations (Al-Naiema et al., 2018; Dai et al., 2019; Schulze et al., 2018). However, the sources and formation mechanisms of these highly oxidized SOAs remain uncertain as the mass spectral features become increasingly similar with atmospheric aging. The diurnal variations of the PMF factors have been analyzed to provide insights into source identification. Driving factors of the diurnal variations of PMF factors include the emission sources, secondary chemical production/loss, boundary layer dynamics, deposition removal processes, and horizontal transport (Janssen et al., 2012; Stefenelli et al., 2019; Takegawa et al., 2006; Zheng et al., 2014). Previous studies conjectured that highly oxidized OA in Houston are relate to daytime photochemistry and mixing from aloft by the boundary layer expansion (Al-Naiema et al., 2018; Bates et al., 2008; Dai et al., 2019). However, the impacts on the diurnal trend by other factors, such as depositional removal and horizontal advection, were not systematically accounted for. For example, given the change of wind direction driven by land/sea breeze, horizontal advection may contribute substantially to the diurnal variations of the aerosol mass concentrations observed in coastal regions.

Most previous studies focused on aerosol properties in the Houston urban area. In comparison, aerosols in the rural areas around Houston are not well understood. Depending on wind direction, the rural areas can experience a range of aerosol conditions, including urban, industrial, marine, and regional background aerosols. The knowledge of aerosol properties and their temporal variations in rural areas allows for an improved understanding of regional aerosol dynamics and representations of aerosols in models. Here, we present the aerosol properties and sources using comprehensive measurements at a rural site southwest of Houston from May to September 2022 during the IOP of the TRACER campaign (Jensen et al., 2022). Different air masses, including those originating from the Gulf of Mexico and strongly influenced by urban emissions, were sampled at the site. The aerosol properties and their temporal variations were characterized for representative air masses. PMF analysis of organic aerosol mass spectra was conducted to identify key OA factors. The sources of OA factors are investigated using (1) the comparative analysis of OA factor mass spectra with those reported in prior studies, (2) the correlation analysis between OA factors and inorganic species (e.g. sulfate, nitrate, ammonium), (3) the dependences of mass concentrations on air mass backward trajectories and local wind patterns, and (4) a box model that includes photochemistry, particle deposition, horizontal and vertical transport. The aerosol properties observed at the rural site are also compared to previous measurements in the Houston region. These analyses help improve our understanding of aerosol properties and processes in rural coastal environments near Houston.

## 2 Methods

## 2.1 Sampling site and measurements

Figure 1. Map showing the location of the rural site (ANC site) during the TRACER campaign (map is from Sentinel-2 cloudless map of the world by EOX).

During TRACER campaign, meteorological parameters, trace gases, and aerosol properties were measured at a rural site (ANC site, 29.37N, 95.75W) in Guy, Texas during the IOP from May 29 to September 29, 2022 (Fig. 1). The ANC site, located on a privately owned farm, is situated approximately 80 kilometers southwest of Houston urban center, 80 kilometers west of the

Houston Ship Channel, and 80 kilometers north of the Gulf of Mexico. The Sam Houston National Forest, which borders the Houston metropolitan area, is about 120 km northeast of the ANC site.

The instruments deployed at the ANC site and the corresponding measurements are described in Table S1. The ANC site had a mixture of Atmospheric Radiation Measurement (ARM)-supported observations (Vaisala automatic weather station, ceilometer, Aerosol Chemical Speciation Monitor (ACSM)) and PI-supported (non-ARM) observations (e.g., Scanning Mobility Particle Sizer (SMPS), Condensation Particle Counter (CPC)). Aerosol and trace gas instruments were housed inside an Aerosol Observing Systems (AOS). The configuration of the AOS is detailed in Uin et al. (2019). The meteorological parameters (surface wind speed, wind direction, air temperature, RH, and air pressure) were measured by an automated weather station (Vaisala). The aerosol inlet was mounted on a mast, 10 meters above the ground level, to minimize the influence of local dust and vehicle emissions. The RH of aerosol samples was reduced to below 20% using a Nafion dryer (Perma Pure) before being introduced into the instruments. The aerosol size distribution ranging from 10 to 500 nm and total particle number concentration were measured by a Scanning Mobility Particle Sizer (SMPS, Model 3082, TSI) and a Condensation Particle Counter (CPC, Model 3772, TSI), respectively. The chemical composition of NR-PM<sub>1</sub> was measured using a Time of Flight - Aerosol Chemical Speciation Monitor (ToF-ACSM, Aerodyne Research) with a standard vaporizer (Fröhlich et al., 2013; Watson, 2017). A cyclone with a cut size of 2.5 µm was installed upstream of the ACSM inlet. Inside ToF-ACSM, the ambient aerosol samples were first focused into a narrow particle beam, passed through a vacuum chamber, and then flash-vaporized at approximately 600 °C. The vaporized species were immediately ionized by 70 eV electron impact, and the resulting ions were analyzed by a time-of-flight mass analyzer. The measured components included organics (Org), sulfate (SO<sub>4</sub>), nitrate (NO<sub>3</sub>), ammonium (NH<sub>4</sub>), and chloride (Chl), and the data have a time resolution of 10 minutes.

## 2.2 PMF analysis

PMF analysis was conducted on the ToF-ACSM mass spectra to identify the key OA components and investigate their sources (Paatero, 1997; Paatero and Tapper, 1994). This methodology operates on the premise that the time series of organic mass spectra can be dissected into several distinct, temporally invariant components. These components, each characterized by their consistent mass spectra, contribute varying quantities of mass concentration to the overall organic signal at each given point in time.

We applied the "rolling PMF" strategy (Canonaco et al., 2021) in this study. To guide the constrained rolling PMF analysis, we first conducted an unconstrained PMF analysis to explore the variability in potential factor profiles and identify suitable candidates for constraints. Multiple solutions with varying numbers of factors were tested, and repeated runs with random seeds were performed to evaluate solution stability. The results were clustered using the k-means methods, and silhouette analysis was used to assess the consistency of profiles within each solution. Based on this evaluation, three factors were selected as reference profiles: HOA, MT-SOA, and isoprene-SOA. These factors were consistently observed across solutions and were chemically interpretable. Using these three factors as constraints, we then performed the rolling PMF analysis. For each rolling window, a random a-value between 0.1 and 0.6 was used to allow flexibility in the factor profiles. We evaluated three different rolling solutions: (1) a 4-factor solution with constrained HOA, MT-SOA and isoprene-SOA factors and one unconstrained OOA factor, (2) a 5-factor solution with the same constrained factors and two unconstrained OOA factors, and (3) a 6-factor solution with the same constrained factors and three unconstrained OOA factors.

We consider the 6-factor solution optimal based on the mass spectral profiles and the correlations of the components with time series for tracer species. The interpretation of these 6 factors will be discussed in Section 3.2. Detailed information on the PMF resolution procedures and solutions comparison is presented in the Supplementary Information (SI) Section S1.

In this study, the O:C ratios of each PMF factor are calculated using the equation from Canagaratna et al. (2015):

O:C ratio = 
$$0.079 + 4.31 \times f_{44}$$
 (1)

where  $f_{44}$  is the ratio of m/z 44 to the total OA signal in the factor mass spectrum. All instruments, including SMPS, CPC, and ToF-ACSM, were deployed from end of the May to end of the September (Table S1).

## 2.3 Classification of air masses and concentration weighted trajectory (CWT) analysis

160

To classify the sampled air masses, we first simulated 24-hour backward trajectories originating at a height of 100 meters above ground level at the ANC site. These trajectories were computed hourly throughout the IOP using the Hybrid Single-Particle Lagrangian Integrated Trajectory (HYSPLIT) model (Stein et al., 2015). The air masses arriving at the site were then classified into three different types according to the backward trajectories following the approach illustrated in Fig. S14 in the SI. Air masses were classified as "marine" if the backward trajectories were over the Gulf of Mexico more than 80% of the 24 hours. Air masses spending less than 80% of the 24 hours over the ocean were considered as either "urban" or "regional", depending on whether the air masses had passed over urban regions. For the air mass classification, the identified urban regions include Corpus Christi (Texas), Houston (Texas), Lafayette (Louisiana), and New Orleans (Louisiana). Aerosols in the "urban" air masses are expected to be substantially influenced by recent anthropogenic emissions. Aerosols in the regional air masses classified here are influenced by continental but not recent urban emissions, therefore they may reflect regional backgrounds. During the IOP from May to September 2022, the predominant air mass type observed at the ANC site is marine, accounting for approximately 60% of all air masses.

The concentration weighted trajectory (CWT) model (Hsu et al., 2003) was used to investigate the potential source areas of major aerosol components (e.g., sulfate, nitrate, organics, ammonium) observed at the ANC site. The analysis domain is chosen between 10° - 60° N and 150° - 50° W based on the farthest distance traveled by 24-hour HYSPLIT backward trajectories. This domain is divided into 5000 grid cells with each grid cell of 1° × 1° in size. A weighted concentration is assigned to each grid cell and is derived by averaging sample concentrations with associated trajectories crossing the grid cell. The CWT values are calculated as follows:

$$CWT_{i,j} = \frac{\sum_{k=1}^{K} c_k \tau_{i,j,k}}{\sum_{k=1}^{K} \tau_{i,j,k}}$$
 (2)

where CWT<sub>i,j</sub> is the CWT value of grid i, j (i: latitude, j: longitude),  $C_k$  is the hourly averaged concentration measured at the ANC site at the start time of trajectory k, K is the total number of hourly back trajectories, and  $\tau_{i,j,k}$  is the number of trajectory points from back trajectory k in grid i, j. Here the trajectory point represents the latitude and longitude at each hour. Therefore, each 24-hour trajectory consists of 24 trajectory points.

## 3 Results and Discussion

170

175

## 3.1 General characteristics of the submicron particles

Figure 2. Time series of NR-PM<sub>1</sub> mass concentrations measured by the ToF-ACSM at the ANC site from May 29 to September 29, 2022.

All time in this paper is local time (UTC-5:00 hours). The marine, urban, and regional air masses are indicated by shades of blue, gray, and light orange, respectively. Also shown are the mass fractions averaged over the four-month IOP. The mass concentration of Chloride represents less than 1% of the NR-PM<sub>1</sub> mass concentration and is neglected.

The campaign average NR-PM<sub>1</sub> mass concentration is 5.2 μg·m<sup>-3</sup>. On average, OA is the largest component and represents 53% of NR-PM<sub>1</sub> mass concentrations. At the ANC site, marine air mass dominated during June, July, and August, while urban air mass was frequently observed in September. On average, NR-PM<sub>1</sub> mass concentration within urban air masses (gray shaded periods in Fig. 2) is approximately 3 times greater than that observed in marine air masses (blue shaded periods in Fig. 2). The difference in NR-PM<sub>1</sub> mass concentration is largely attributed to strong anthropogenic emissions in the Houston's urban area (Bahreini et al., 2009; Brock et al., 2003; Brown et al., 2013). In urban air masses, OA dominates and represents 66% of NR-PM<sub>1</sub> mass concentration (Fig. 3f). In contrast, the mass fraction of sulfate becomes comparable to OA in marine air masses (Fig. 3d), likely due to shipping emissions in the Gulf of Mexico (Schulze et al., 2018; Sullivan et al., 2013; Wallace et al., 2018; Zhou et al., 2023).

Figure 3. The 1<sup>st</sup> row (a-c): averaged aerosol size distributions during IOP in (a) Marine, (b) Regional, and (c) Urban air masses. The solid line represents the median values, and the error bars indicate the 25<sup>th</sup> and 75<sup>th</sup> percentiles. The 2<sup>nd</sup> row (d-f): averaged NR-PM<sub>1</sub> mass concentrations and fractions in (d) Marine, (e) Regional, and (f) Urban air masses.

The aerosol size distributions in the three air mass types are shown in Fig. 3 and Table 1. The average total particle number concentration in the urban air mass is 3 times that in marine air mass. Aerosol size distribution in marine air masses shows a bimodal spectral shape (Fig. 3a), a common feature attributed to in-cloud processing (Gong et al., 2023; Hoppel et al., 1986). In contrast, urban air masses exhibit a unimodal aerosol size distribution, consistent with previous measurements in urban areas (Chen et al., 2022; Dall'Osto et al., 2012; Hussein et al., 2004). The observed modal diameters in marine (60 and 150 nm) and urban (65 nm) air masses are consistent with previous observations in Houston (Levy et al., 2013; Schwarz et al., 2008). The diurnal variations of aerosol size distribution reveal elevated concentrations of particles smaller than 30 nm around noon in both marine and urban air masses (Fig. S15). The elevated nucleation mode particle concentrations are consistent with previous field observations in the Houston region (Russell et al., 2004, Levy et al., 2013) and are attributed to new particle formation (Fan et al., 2006). New particle formation around noon is commonly observed in urban environments (Brines et al., 2015; Minguillón et al., 2015; Reche et al., 2011) and is likely due to elevated gas phase concentrations of sulfuric acid and low-volatility organic compounds resulting from photochemistry.

Table 1. Averages and standard deviations of number concentrations for different aerosol modes, total particle number concentration, mass concentrations of NR-PM<sub>1</sub> species and PMF OA factors in marine, regional, and urban air masses observed at the ANC site during IOP.

|                                                                           |                                                                           | Marine          | Regional        | Urban           |
|---------------------------------------------------------------------------|---------------------------------------------------------------------------|-----------------|-----------------|-----------------|
| Mode number concentration                                                 | Nucleation Mode ( $D_p \le 20 \text{ nm}$ ) (× $10^3 \text{ cm}^{-3}$ )   | $0.30\pm0.82$   | $0.30\pm0.82$   | $0.94 \pm 3.59$ |
|                                                                           | Aitken Mode (20 nm $<$ $D_p<$ 100 nm) ( $\times$ 10 $^3$ cm $^{-3}$ )     | $0.89 \pm 0.81$ | $1.55\pm7.83$   | $4.84\pm8.58$   |
|                                                                           | Accumulation Mode ( $D_p >= 100 \text{ nm}$ ) (× $10^3 \text{ cm}^{-3}$ ) | $0.46\pm0.35$   | $0.57 \pm 0.98$ | $1.27\pm0.64$   |
| Total particle number concentration (× 10 <sup>3</sup> cm <sup>-3</sup> ) |                                                                           | $2.21 \pm 2.72$ | $2.71 \pm 2.44$ | $6.87 \pm 7.83$ |
| Mass<br>concentration of<br>NR-PM <sub>1</sub> species                    | Organics (Org) (µg·m <sup>-3</sup> )                                      | $1.42 \pm 1.94$ | $2.17 \pm 2.37$ | $6.58 \pm 3.62$ |
|                                                                           | Sulphate (SO <sub>4</sub> ) ( $\mu g \cdot m^{-3}$ )                      | $1.47 \pm 0.92$ | $1.39\pm1.08$   | $2.15\pm1.61$   |
|                                                                           | Ammonium (NH <sub>4</sub> ) (μg·m <sup>-3</sup> )                         | $0.50\pm0.31$   | $0.50 \pm 0.47$ | $0.74 \pm 0.48$ |
|                                                                           | Nitrate (NO <sub>3</sub> ) (μg·m <sup>-3</sup> )                          | $0.16\pm0.22$   | $0.29 \pm 0.67$ | $0.49 \pm 0.43$ |
| Mass concentration of PMF OA factors                                      | HOA (μg·m <sup>-3</sup> )                                                 | $0.07\pm0.14$   | $0.12 \pm 0.22$ | $0.24 \pm 0.22$ |
|                                                                           | MT-SOA (μg·m <sup>-3</sup> )                                              | $0.14 \pm 0.22$ | $0.26 \pm 0.33$ | $0.82 \pm 0.69$ |
|                                                                           | isoprene-SOA (μg·m <sup>-3</sup> )                                        | $0.23 \pm 0.44$ | $0.32 \pm 0.44$ | $1.09 \pm 0.74$ |
|                                                                           | shipping-OOA ( $\mu g \cdot m^{-3}$ )                                     | $0.15 \pm 0.26$ | $0.20 \pm 0.30$ | $0.84 \pm 0.69$ |
|                                                                           | OOA1 (μg·m <sup>-3</sup> )                                                | $0.30 \pm 0.50$ | $0.48 \pm 0.61$ | $1.62 \pm 1.11$ |
|                                                                           | OOA2 (μg·m <sup>-3</sup> )                                                | $0.47 \pm 0.59$ | $0.66 \pm 0.62$ | $1.50 \pm 0.84$ |

To investigate the origins of different aerosol components, we examined the correlation between each chemical component, conducted the CWT analysis, and examined the dependence of component concentrations on local wind speed and direction. Strong correlations were observed between nitrate and organics ( $R^2 = 0.54$ ) and between sulfate and ammonium ( $R^2 = 0.83$ ) (Table S2). Both the CWT analysis and wind-rose plots indicate that elevated organic mass concentration in air masses passing over urban and forested areas (Fig. 4 and Fig. 1). In contrast, sulfate has contributions from both urban area and Gulf of Mexico, similar to findings in previous studies (Al-Naiema et al., 2018; Cleveland et al., 2012; Dai et al., 2019; Schulze et al., 2018). Ammonium exhibits a similar spatial distribution to sulfate. However, because marine emissions are unlikely a major source of ammonia, the similarity likely reflects the formation of ammonium sulfate or bisulfate through atmospheric neutralization processes involving anthropogenic sulfate and terrestrial ammonia (Schiferl et al., 2014; Weber et al., 2016). The similar CWT and wind-rose patterns for nitrate and organics suggest that nitrate may contain organic nitrate (Fig. 4). Unfortunately, the resolution of the ACSM deployed at the ANC site is insufficient to differentiate organic and inorganic nitrates.

Figure 4. The  $1^{st}$  row: CWT analysis of mass concentrations of NR-PM<sub>1</sub> species. The  $2^{nd}$  row: Wind-rose plots showing the variations of mass concentrations of NR-PM<sub>1</sub> species with wind direction and speed.

# 3.2 Positive Matrix Factorization (PMF) analysis of OA

Figure 5. (a-f) Mass spectra of PMF OA factors and (g-l) Time series of OA factors. The correlations between OA factors and relevant species are also shown (h, i).

We applied PMF analysis to classify OA into 6 factors, including hydrocarbon-like OA (HOA), monoterpene-derived secondary OA (MT-SOA), isoprene-derived secondary OA (isoprene-SOA), shipping emission related OOA (shipping-OOA), oxidized OA-1 (OOA1), and oxidized OA-2 (OOA2) (Fig. 5). On average, these factors contribute 6%, 12%, 17%, 12%, 25%, 28%, respectively, to the OA mass concentrations during the IOP (Fig. 6a). In the following sections, we examine the potential sources of the OA factors by comparing the mass spectra of the factors with those reported in previous studies (Jeon et al., 2023) and by analyzing the correlations between OA factors and inorganic species, the air mass backward trajectories, and the dependence of OA factor mass concentrations on wind direction and speed.

220

Figure 6. (a) Mass fractions of PMF OA factors over IOP. (b) Diurnal variations of OA factors mass concentrations during the IOP.

## 3.2.1 HOA, MT-SOA, and isoprene-SOA

The HOA profile is dominated by fragments of aliphatic hydrocarbons, including m/z 41 ( $C_3H_5^+$ ), 55 ( $C_4H_7^+$ ), 57 ( $C_4H_9^+$ ), 69 ( $C_5H_9^+$ ), and 71 ( $C_5H_{11}^+$ ) (Fig. 5a). These chemical formulas are based on measurements of high-resolution Aerosol Mass Spectrometers (AMS) from previous studies. The HOA mass spectrum in this study exhibits a strong correlation ( $R^2$ =0.83) with the spectrum of HOA factors identified in Mohr et al. (2012) and Docherty et al. (2011) (Table S3). The lower mass concentration of HOA during daytime (Fig. 6b) is attributed to increased planetary boundary layer height (PBLH) and the negligible contribution of secondary species. Fig. S17 shows that HOA mass concentration becomes elevated when the wind is from the northeast, i.e., the direction of the Houston Ship Channel, suggesting that shipping emissions likely represent a major source of HOA at the site.

Among all 6 OA factors, MT-SOA has the highest  $f_{91}$  (i.e., ratio of m/z 91 to total OA signal in the factor mass spectrum, Fig. 5b and Fig. S18b). High  $f_{91}$  value is characteristic of SOA from the oxidation of monoterpenes, as shown by previous laboratory studies (Boyd et al., 2015; He et al., 2021; Takeuchi et al., 2022). The mass spectrum of the MT-SOA factor closely matches those of laboratory SOA produced from the nitrate radical oxidation of limonene (Boyd et al., 2015) and a mixture of α-pinene and limonene (Takeuchi et al., 2022), with the  $R^2$  values of 0.90 and 0.92, respectively (Table S3). In addition, MT-SOA correlates with NO<sub>3</sub> with  $R^2$ =0.58 (Fig. 5h and Table S4). A similar correlation has been observed in previous field studies (Budisulistiorini et al., 2015; Hao et al., 2014; Kiendler-Scharr et al., 2016; Xu et al., 2015) and suggests a substantial contribution of organic nitrates to MT-SOA. Elevated MT-SOA mass concentration was observed with north and northeast winds (Fig. S17) from the Sam Houston National Forest, where there are strong emissions of monoterpenes (Brown et al., 2013). We note that OA with a high  $f_{91}$  can be associated with aged biomass-burning OA (BBOA) (Robinson et al., 2011). However, similar to observations in the southeastern U.S. (Budisulistiorini et al., 2015, 2016), MT-SOA in this study does not show strong signals at m/z 60 or 73, which are characteristic of levoglucosan (Alfarra et al., 2007; Schneider et al., 2006). Additionally, signals at m/z 18, 29, and 44, which are used as tracers for BBOA in some studies (Bougiatioti et al., 2014), are also negligible for the MT-SOA. Therefore, we attribute MT-SOA to SOA from the oxidation of monoterpenes.

As high  $f_{82}$  is characteristic of isoprene SOA from the reactive uptake of isoprene epoxydiols in the presence of acidic sulfate particles (Hu et al., 2015; Lin et al., 2011; Lv et al., 2019; Riva et al., 2016; Xu et al., 2015), the OA factor with the highest  $f_{82}$  is denoted as isoprene-SOA (Fig. 5c and Fig. S5a). The mass spectrum of isoprene-SOA factor in this study agrees well with those of the Fac82 factor observed in the Borneo rainforest (Robinson et al., 2011) ( $R^2 = 0.93$ ) and IEPOX-SOA factor during the SOAS

campaign (Hu et al., 2015) ( $R^2 = 0.88$ ). The dominant land cover in the immediate vicinity of the ANC site is grassland. Global estimates suggest that grasses and herbaceous plants emit much less isoprene than trees, contributing less than 4% of the total annual global isoprene emissions (Bai et al., 2006). Isoprene-SOA shows elevated mass concentration when the wind is from the northeast, the direction of Sam Houston National Forest (Fig. S17a). Therefore, while isoprene is emitted from grasslands nearby, forest emissions are likely the dominant isoprene source for the isoprene-SOA observed at the site. The mass concentrations of isoprene-SOA factor and sulfate are positively correlated with an  $R^2$  value of 0.36 (Fig. 5i and Table S4). This  $R^2$  value is comparable to that observed in the Amazon rainforest ( $R^2 = 0.37$ ; de Sá et al., 2017), but slightly lower than those reported in the southeastern U.S. (0.48 – 0.6) (Budisulistiorini et al., 2013, 2015; Hu et al., 2015; Xu et al., 2015). Unlike previous studies, where sampling sites were located in forested areas with broadly distributed isoprene emissions, isoprene at our ANC site primarily originates from the Sam Houston National Forest to the broad north, while a major source of sulfate is Houston Ship Channel to the northeast (Fig. 4). The spatial separation of isoprene and sulfate sources may contribute to the relatively lower  $R^2$  value in this study. This spatial separation is further supported by wind-dependent trends of isoprene-SOA, with elevated concentrations occurring when winds are from the north and northeast (Fig. S17a).

Figure 7. A shipping-OOA pollution event from local time 4:00 to 8:00 on September 19, 2022. Time-series of OA factors (a) mass concentrations and (b) mass fractions. (c) f55 vs. f57 of PMF factors from this and prior studies. (d, e) Backward trajectories originating from 100 m above ground at the ANC site on 19 Sep. 2022, colored according to shipping-OOA mass concentration; the area marked by the blue box in (d) is shown in (e).

# 3.2.2 Shipping-OOA

275

260

265

One OOA factor exhibits the highest  $f_{55}$  among all OOA factors, with a  $f_{55}/f_{57}$  ratio greater than 2 (Fig. 7c). OA factors with similar mass spectra were observed in previous studies in the Houston area and attributed to OA from cooking emissions (Al-Naiema et

al., 2018; Wallace et al., 2018). The attribution to cooking emissions was mainly based on the high  $f_{55}$  signal and  $f_{55}/f_{57}$  ratio (Mohr et al. 2012). However, in this study, as well as in the previous studies in Houston (Al-Naiema et al., 2018; Wallace et al., 2018), the diurnal variations of the factors with high  $f_{55}$  signals and  $f_{55}/f_{57}$  do not show elevated mass concentrations around meal times, when cooking emissions peak (Fig. 6b). In addition, there are no major cooking activities near the ANC site. Cooking is not the only source for OA with high  $f_{55}$  signals and  $f_{55}/f_{57}$  values above 2. OA from shipping emissions could have similar features (Schulze et al., 2018). The mass spectrum of this factor agrees well with that of organics during periods of heavy shipping emissions reported by Schulze et al. (2018), with an  $R^2$  value of 0.91 (Fig. S20), suggesting that this factor is likely associated with shipping emissions instead. Observations from 4:00 to 8:00 on September 19 provide additional evidence that relates this factor to shipping emissions. At 4:00, the concentration of this factor increased (Figs. 7a and 7b) as the air mass started to pass over Freeport, a ship port (Figs. 7d and e). The concentration remained elevated until 8:00 when the air mass trajectory began to move away from the port area. Collectively, we identify and refer to this factor as shipping-OOA. High shipping-OOA mass concentration was observed when the wind is from the northeast, i.e., the direction of Houston Ship Channel (Fig. S17a), supporting that shipping emissions are the dominant source of shipping-OOA. In addition, the CWT analysis shows shipping-OOA hotspots over Freeport for the marine air mass (Fig. S21). During the IOP, shipping-OOA observed at the ANC site accounts for 12% of the total OA (Fig. 6a), compared with 8.6-32% in previous studies (Al-Naiema et al., 2018; Wallace et al., 2018). These earlier studies may have underestimated the contribution of shipping emissions while overestimating the contribution of cooking emissions to the OA in Houston.

## **3.2.3 OOA1 and OOA2**

Two oxygenated OA factors with different O:C ratios (1.05 and 1.36 for OOA1 and OOA2, respectively) were identified, collectively accounting for 53% of OA mass concentration (Fig. 6a). As the OA mass spectra become increasingly similar through the aging process, separating OOA factors and identifying their precursors and sources is more challenging compared to other OA factors (Hass-Mitchell et al., 2024; Ng et al., 2011; Zhang et al., 2011). OOA1 accounts for 25% of OA mass concentration and has an O:C ratio of 1.05, between those of shipping-OOA and OOA2. The O:C ratio falls within the range of highly oxidized OAs observed in urban areas (Aiken et al., 2008; Al-Naiema et al., 2018). The much higher OOA1 mass concentration in urban air masses than in the marine air masses suggests the precursors are mostly from emissions over the land (Table 1). The CWT analysis for urban air mass shows OOA1 hotspots over downtown Houston and the Houston Ship Channel, suggesting the influence from local urban emissions. In contrast, OOA2 displays a more spatially uniform distribution, indicating a more regional source (Fig. S22). This interpretation is further supported by the variations of OOA1 and OOA2 mass concentrations with wind direction (Fig. S17).

OOA2 has the highest oxidation level and represents the largest fraction of OA mass concentration (28%, Fig. 6a). The mass spectrum of OOA2 matches those of highly aged OA in the literature (Lanz et al., 2007; Zhang et al., 2005). Wind-rose plots show that the OOA2 mass concentration in either marine or urban air mass is largely independent of the wind direction (Fig. S17), suggesting minimal influence from local emissions. At the ANC site, the mass concentration of OOA2 shows an increase starting in the morning (9:00) followed by a decrease starting in the early afternoon (14:00~15:00) (Fig. 6b). Highly oxidized OA factors with similar diurnal variations were commonly observed in the Houston area during previous studies (Al-Naiema et al., 2018; Bates et al., 2018; Dai et al., 2019; Schulze et al., 2018; Wallace et al., 2018). The midday elevated mass concentration of the highly oxidized OA factors was previously attributed to SOA formation driven by photochemical reactions and mixing from aloft (Al-Naiema et al., 2018; Bates et al., 2008; Dai et al., 2019). Here we find that the diurnal trend of OOA2 at the ANC site is

primarily controlled by the change in air masses. For each individual air mass type (i.e., marine or urban), the OOA2 mass 315 concentration largely remains constant throughout the day (Fig. 8a), in contrast to the diurnal variation of OOA2 in all air masses (referred to as "unseparated air masses" thereafter). To investigate the influence of air masses on the diurnal variation of OOA2 mass concentration, we first examined the time of air mass spent over the land during the 24 hours before arriving at the site using backward trajectories. The time of air mass spent over the land shows a similar diurnal variation as OOA2 mass concentration (Fig. 320 8b). The percentage of urban air mass observed at the site also shows a midday enhancement (Fig. 8c). This enhancement is likely due to diurnal variation of wind direction under the influence of land breezes and sea/bay breezes in Houston. Specifically, the wind alternates between northerly and southerly directions, with northerly winds prevailing in the evening with lower wind speed (land breezes) and southerly winds dominating during the daytime with higher wind speed (sea/bay breezes) (Fig. S25a). We calculated the backward trajectories of air masses arriving at the site in the early afternoon (i.e., 13:00) and evening (i.e., 21:00) 325 using wind direction and speed averaged over days during the IOP on which air mass change was observed (Fig. S25c and d). The result clearly shows that the air mass observed at 13:00 spends a higher fraction of time over the land (and is more likely to be influenced by urban emissions) than the air mass observed at 21:00 (Fig. S25b). Given the higher OOA2 mass concentration in the urban air mass (Table 1), the elevated urban air mass fraction leads to enhanced OOA2 mass concentration at the ANC site midday.

To further investigate the processes driving the diurnal variation of OOA2, we employed a box model as described in Chen et al. (2021). The model considers direct emission, chemical reaction, depositional loss, horizontal advection, and vertical transport, and the temporal variation of the concentration of species i, ( $c_i$ ) is given by:

$$\frac{\mathrm{d}c_i}{\mathrm{d}t} = \frac{q_i}{H(t)} + R_i - \frac{V_{d_i}}{H(t)}c_i + \frac{u}{\Delta x}(c_i^0 - c_i) + \frac{1}{H(t)}\frac{\mathrm{d}H}{\mathrm{d}t}|_{\mathrm{d}H/\mathrm{d}t > 0}(c_i^a - c_i)$$
(3)

where H(t) is the PBLH and is derived from ceilometer measurements,  $q_i$  is the emission rate,  $R_i$  is the chemical production and loss rate,  $V_{d_i}$  is the deposition velocity, u is the wind speed in the constant  $\Delta x$  direction,  $c_i^0$  is the background concentration of species i,  $c_i^a$  is the concentration of species i aloft.

The following key assumptions were applied in this model for marine and urban air masses. First, there is no direct emission of OOA2 because it is an aged SOA factor. Second, the net effect of horizontal advection is negligible within the same air mass type. This assumption is based on the weak dependence of OOA2 mass concentration with wind direction for the same air mass type (Fig. S17). Third, the chemical production term includes the oxidation of isoprene-SOA and OOA1. The further oxidation of OOA2 represents a sink. With these assumptions, Eq. (3) can be written as:

$$\frac{dm_{\text{OOA2}}}{dt} = (k_1 m_{\text{isoprene-SO}} + k_2 m_{\text{OOA1}} - k_3 m_{\text{OOA2}}) m_{\text{OH}} - \frac{V_{\text{d}}}{H(t)} m_{\text{OOA2}} + \frac{1}{H(t)} \frac{dH}{dt} |_{dH/dt > 0} (m_{\text{OOA2}}{}^a - m_{\text{OOA2}})$$
(4)

The reaction rates  $k_1$ ,  $k_2$ , and  $k_3$  were set to  $5 \times 10^{-12}$ ,  $1 \times 10^{-12}$ , and  $1 \times 10^{-13}$  cm<sup>3</sup> molecule<sup>-1</sup> s<sup>-1</sup>, respectively (Chen et al., 2021). Sensitivity tests for these rate constants were conducted, with  $k_1$ ,  $k_2$ , and  $k_3$  reduced by 50% (Fig. S23) and increased by 50% (Fig. S24). The results indicate that these changes have minimal impact on the overall model outcomes. The OH was not measured at the ANC site during the IOP. Li et al. (2012) suggested that OH concentrations in rural areas around Houston are approximately 2–10 times lower than those in downtown. We divided the OH concentrations measured in downtown Houston (Ren et al., 2013) by a factor of 5 to estimate the OH concentration at the rural ANC site.  $V_d$  was calculated based on the diurnal variation of aerosol

volume average diameter, wind speed, and temperature (Emerson et al., 2020). The aloft OOA2 concentration  $(m_{00A2}{}^a)$  was assumed to be constant and derived by fitting the diurnal variation predicted by Eq. (4) to the measured. The derived values of  $m_{00A2}{}^a$  are 0.43  $\mu$ g·m<sup>-3</sup> and 1.45  $\mu$ g·m<sup>-3</sup> for marine and urban air masses, respectively.

The model successfully captures the observed OOA2 diurnal trends in both marine and urban air masses (Figs. 9a and 9b). Figs. 9d and e show the diurnal variations of the overall change rate of OOA2 mass concentration and contributions from chemical production/loss, deposition, and mixing from aloft for marine and urban air masses. "Mixing from aloft" represents the vertical transport of OOA2 between the boundary layer and the free troposphere as the PBLH changes. The PBL is shallow at night and grows during the daytime (Fig. S16). When the PBLH increases, free troposphere air entrains into the boundary layer, leading to dilution of OOA2 if the concentration aloft is low or enrichment if the concentration aloft is high. In contrast, when the PBLH decreases, there is no mixing between the boundary layer and the free troposphere. Therefore, the decreasing PBLH has no impact on OOA2 concentration at the surface. "Chemical Production/Loss" refers to the production and loss of OOA2 through OH oxidation. "Deposition Loss" represents the removal of OOA2 via dry deposition.

Figs. 9d and e show the contributions of each process to the change in OOA2 mass concentration for marine and urban air masses, respectively. Overall, chemical reactions lead to an increase in OOA2 mass concentration throughout the day, with the highest production rate around noon time due to elevated OH concentrations. Deposition loss is higher at night due to the shallower PBLH. The mixing from aloft influences OOA2 concentration from early morning to early afternoon when the PBLH increases. In the late afternoon, the PBLH starts to decrease, but this process does not directly affect the OOA2 concentration. The change rates of OOA2 concentration due to these three processes are quite small, less than 0.02 μg·m<sup>-3</sup> h<sup>-1</sup> for marine air mass and 0.04 μg·m<sup>-3</sup> h<sup>-1</sup> for urban air mass. Combined, these processes result in a negligible net change rate, and therefore, OOA2 concentrations are largely constant throughout the day for both marine and urban air masses. These box model results further support that the OOA2 is highly aged, minimally influenced by local emissions, and represents a uniform background within the same air mass type.

We also modeled the diurnal variation of OOA2 mass concentration for unseparated air masses using the 1-D box model with the same assumptions described above (Figs. 9c and f). We initially assumed a constant OOA2 concentration aloft, but the model was not able to capture the observed diurnal variation. We then derived an altitude-dependent OOA2 aloft concentration by correlating the observed OOA2 concentration with PBLH (Fig. S19), as in Chen et al. (2021). While an altitude-dependent OOA2 improves the agreement between the model results and observations to some extent, the 1-D model still fails to capture the diurnal trend of OOA2 in unseparated air masses (Fig. 9c). First, the modeled increase of OOA2 concentration during daytime is much more gradual than the observed. Second, at around 15:00 local time, the observed OOA2 concentrations begin to decrease, whereas the simulated concentrations remain constant. These discrepancies are probably due to that the 1-D model neglects the impact of horizontal advections. For the same air mass type, the effect of horizontal advection is likely negligible, as supported by the observation that OOA2 mass concentration is largely wind-direction independent. However, given the contrasting OOA2 mass concentrations in different air mass types (Table 1), the impact of horizontal advection can be substantial when the air mass type observed at the ANC site shifts. As discussed earlier, the air masses observed during midday tend to have spent more time over land compared to those observed in morning and evening, and they are more likely influenced by urban emissions (i.e., a higher percentage of the urban air mass, Fig. 8b, and c). Given the higher  $m_{OOA2}$  in the urban air mass, the elevated  $m_{OOA2}$  during midday is attributed to the higher percentage of urban air masses arriving at the site. Previous studies have observed similar diurnal

variations of highly oxidized OA in the Houston area (Al-Naiema et al., 2018; Bates et al., 2008; Dai et al., 2019), and attributed the variations to secondary aerosol formation by photochemical reactions and mixing from aloft. Our analysis indicates that the variation observed at the ANC site is likely dominated by the shifting in air mass (i.e., aerosol sources), and the influence of secondary formation and mixing from aloft is relatively minor.

Figure 8. (a) Diurnal variations of observed OOA2 mass concentrations in marine, urban, and unseparated air mass; (b) Diurnal variations of observed OOA2 mass concentrations and the time spent by the air mass over land during the 24 hours before arriving at the site; (c) Diurnal variations of observed OOA2 mass concentration and the percentage of the urban air mass. The shaded areas in A and B represent  $\pm$  one standard deviation.

Figure 9. 1-D Box model results. (a, b, c) Diurnal variations of observed and modeled OOA2 mass concentrations in (a) marine, (b) urban, and (c) unseparated air masses; (d, e, f) Simulated contributions from different processes (mixing from aloft, chemical production/loss, deposition loss) and the net change rate of OOA2 mass concentration within the PBLH in the (d) marine, (e) urban, and (f) unseparated air masses.

# 4 Comparison with other studies

Fig. 10 and Table S5 summarize the NR-PM<sub>1</sub> mass concentrations and mass fractions at various locations in the great Houston region, based on this and previous studies. Caution is needed when comparing these results as these studies were conducted in different months over several years. In general, the total aerosol mass concentration in the greater Houston area influenced by urban air masses (10.8 μg·m<sup>-3</sup> at Manchester St.; 10.9 μg·m<sup>-3</sup> at the University of Houston; 9.8 μg·m<sup>-3</sup> within the urban air mass at Southwest of Galveston; 9.96 μg·m<sup>-3</sup> within the urban air mass at Guy) is approximately three times higher than that influenced by marine air masses (3.82 μg·m<sup>-3</sup> within the marine air mass at Southwest of Galveston; 3.55 μg·m<sup>-3</sup> within the marine air mass at Guy) (Table S5). The major local aerosol sources in the greater Houston area include the industrial and traffic emissions in and

around the Houston urban area and shipping emissions near the coastal line. To visualize their impacts on major aerosol components, we generate heatmaps illustrating the variations of sulfate, SOA, HOA mass concentrations and sulfate mass fractions with distances to the urban center and the coastal line (Fig. 10b, d, e, and c). Here, SOA is defined as the sum of all PMF factors except for the primary organic aerosol factors. The urban center is defined as the University of Houston.

Sulfate exhibits higher mass concentrations near both the urban center and coastal line (Fig. 10b). Cleveland et al. (2012) reported the highest sulfate concentration of 4.1 µg·m³ (Fig. 10b and Table S5) at the University of Houston. The second-highest sulfate concentration, 2.5 µg·m³, was observed at a sampling site approximately 6.5 kilometers from the University of Houston and surrounded by industrial and petrochemical complexes (Wallace et al., 2018). Due to the proximity of the two sampling locations, the 1.64-fold difference in sulfate concentrations may be due to temporal variations rather than spatial differences, suggesting a decline in anthropogenic emissions in Houston. At the southwest of Galveston, near the coastal line, sulfate concentration was observed as 2.4 µg·m³ in marine air mass (Schulze et al., 2018), comparable to that in downtown Houston. While sulfate concentrations in both areas are similar, the sulfate mass fraction near the coastline is substantially higher (63%) than that near the urban center (23%) (Fig. 10c), suggesting different aerosol sources and processes. In downtown Houston, the primary source of sulfate is refinery emissions (Wallace et al., 2018), whereas the sulfate near the coastal regions is mainly from the shipping emission (Schulze et al., 2018; Zhou et al., 2023). The lower sulfate fraction near the urban center is largely driven by higher SOA mass concentration near the urban center (Fig. 10d), which is attributed to the abundant VOCs (Bahreini et al., 2009) and oxidizing agents (such as ozone, OH, and nitrate radicals) (Paraschiv et al., 2020) from industrial and traffic emissions. HOA exhibits higher mass concentrations near both the urban center and coastal line (Fig. 10e), suggesting significant contributions from primary emissions related to fossil fuel combustion, including vehicular traffic, industrial activities, and shipping emissions.

The above comparison shows that the aerosol mass concentrations and compositions observed in urban and marine air masses at the ANC site are consistent with earlier results. Together, the measurements at the ANC site and other locations show that the industrial and traffic emissions in the urban center, as well as shipping emissions along the coastal line, are among the important aerosol sources in the Houston region, including the rural area where the ANC site is located.

Figure 10. (a) Sample locations and aerosol compositions measured by ACSM/AMS during this and previous field studies in the Houston region (© Google Maps 2022). Pie charts show the average mass fractions of NR-PM<sub>1</sub> species and PMF resolved OA factors. (b) Mass concentration of sulfate, (c) Mass fraction of sulfate, (d) Mass concentration of SOA, and (e) Mass concentration of HOA. The mass concentrations and fractions in (b, c, d, e) is from marine air mass for locations Guy and HSC.

<sup>\*</sup>Explanation of the abbreviation: HSC: Houston Ship Channel; UoH: University of Houston; Manchester St./M. St.: Manchester Street; SL: Sugar Land; SW of Galveston/SWG: Southwest of Galveston;

#### Conclusion

In this study, we present a comprehensive analysis of aerosol properties measured at a coastal-rural site (i.e., ANC site) near Houston, Texas, during the TRACER campaign. Based on 24-hour backward trajectories, air masses arriving at the site are classified into three different types: marine air mass from the Gulf, urban air mass influenced by urban emissions, and regional air mass. Marine air masses typically exhibit bimodal aerosol size distribution due to cloud processing and have the lowest particle number and PM<sub>1</sub> mass concentrations among all three air mass types, whereas urban air masses show the highest number and PM<sub>1</sub> mass concentrations. On average, particle number and mass concentrations in urban air masses are three times higher than those in marine air masses.

Using PMF analysis on aerosol mass spectra, we identified 6 OA factors, including hydrocarbon-like OA (HOA), OA from the oxidation of monoterpenes (MT-SOA), OA from the reactive uptake of isoprene epoxydiols by acidic sulfate particles (isoprene-SOA), oxygenated OA arising from shipping emissions (shipping-OOA), and two oxygenated OA factors with high O:C ratios (OOA1 and OOA2). On average, these factors contribute 6%, 12%, 17%, 12%, 25%, 28%, respectively, to the OA mass concentration during the IOP. The dependence of HOA mass concentration on wind direction suggests that shipping emissions as its major source. Based on mass spectra signature and wind direction dependence, MT-SOA and isoprene-SOA are attributed to the oxidation of monoterpenes emitted from Sam Houston National Forest and reactive uptake of isoprene epoxydiols in the presence of acidic sulfate particles, respectively. Shipping-OOA factor has the highest  $f_{55}$  among all OOA factors, with an  $f_{55}/f_{57}$  ratio exceeding 2. Our analysis indicates that this factor is likely associated with shipping emissions rather than cooking emissions suggested by previous studies.

Collectively, two oxygenated OA factors with high O:C ratios (1.05 and 1.36 for OOA1 and OOA2, respectively) accounting for 53% of OA mass concentration are observed at the site. The O:C ratios of both OOA1 and OOA2 fall within the range of highly oxidized OA typically observed in urban areas. As the OA mass spectra become increasingly similar through the aging process, identifying specific precursors and sources of OOA1 and OOA2 proves challenging. The CWT analysis for the urban air mass indicates potential contribution of local emissions in downtown Houston and the Houston Ship Channel areas to OOA1. The weak dependence of mass concentrations on wind direction in marine or urban air mass suggests that local sources have relatively minor contributions to OOA2. OOA2 has the highest oxidation level and represents the largest fraction of OA mass (28%). At the ANC site, the OOA2 mass concentration peaks midday (i.e., ~ 11:00 to ~ 16:00). Highly oxidized OA factors with similar diurnal variations have been commonly observed in the Houston area during prior studies, where the midday peak was attributed to SOA formation driven by photochemical reactions and mixing from aloft. Utilizing air mass backward trajectories and a 1-D box model, we demonstrate that the diurnal trend of OOA2 at the ANC site is predominantly influenced by the change of air masses instead. Both the duration of air masses over land and the fraction of urban air mass observed at the site show a midday enhancement, which is likely due to the diurnal variation of wind direction under the influence of land breezes and sea/bay breezes in the Houston area. Given the higher OOA2 mass concentration in urban air masses, the high urban air mass fraction in midday leads to elevated OOA2 mass concentration at the ANC site.

The aerosol mass concentrations and compositions in urban and marine air masses observed at the ANC site are consistent with results from previous studies in the Houston region. Together, the measurements at the ANC site and other locations consistently show that shipping emissions along the coastal line, as well as the industrial emissions and traffic emissions in the urban center are

among the important aerosol sources in the Houston region, including at the rural area where the ANC site is located. This study quantifies aerosol properties in representative air masses, identifies the major sources of OA in the Houston region, and highlights the impacts of emissions, atmospheric chemistry, and meteorology on aerosol properties in the coastal-rural environment.

Code and data availability. TRACER observational datasets are available at <a href="https://www.arm.gov/data/">https://www.arm.gov/data/</a>. HYSPLIT data are accessible through the NOAA READY website (<a href="https://www.ready.noaa.gov/HYSPLIT.php">https://www.arm.gov/data/</a>. HYSPLIT data are accessible through the NOAA READY website (<a href="https://www.ready.noaa.gov/HYSPLIT.php">https://www.arm.gov/data/</a>. HYSPLIT data are accessible through the NOAA READY website (<a href="https://www.ready.noaa.gov/HYSPLIT.php">https://www.arm.gov/data/</a>. HYSPLIT data are accessible through the NOAA READY website (<a href="https://www.ready.noaa.gov/HYSPLIT.php">https://www.ready.noaa.gov/HYSPLIT.php</a>, NOAA Air Resources Laboratory, 2022). The code used to generate the figures is available upon request.

490 *Author contributions*. JW, LX, and JL designed the study. JL, JZ, XG, SS, CK, AS, and MZ carried out the measurements. MZ resolved the PMF factors. JL, JW, and LX led the data analysis and preparation of manuscript, with contributions from all authors.

Acknowledgements. We thank Daniel Bahrt and Ana Gabriela Pessoa, technicians from Atmospheric Radiation Measurement (ARM) Climate Research Facility, for their help on the facilities and instruments maintenance during IOP.

*Financial support.* This research was conducted with funding from the Atmospheric Radiation Measurement (ARM) Climate Research Facility (Office of Biological and Environmental Research of the U.S. Department of Energy, Climate and Environmental Science Division) under contracts DE-SC0021017.

Completing interests. The authors declare that they have no conflict of interest.

495

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
