# Peer review of "The sources and diurnal variations of submicron aerosols in a coastal-rural environment near Houston, US"

_EGUsphere, 2025_

## Author Comment (AC1)

**Response to RC1:**

We thank the reviewers for their thoughtful and constructive comments. Please find below the response to each comment or question, including notations of improvements to the manuscript. Reviewer comments are in blue fonts and responses are in black. Changes to the manuscript are indented. As the reviewer suggested, we renamed the PMF factor "91FAC" to "MT-SOA" to better reflect its source and chemical identity. Similarly, "OOA1" was renamed to "shipping-OOA" to avoid the confusion of classifying it as either POA or SOA. Consequently, "OOA2" and "OOA3" were renamed as "OOA1" and "OOA2" in the revised main manuscript and supplementary information. We retained the original factor names in the response to reviewers to ensure consistency with the terminology used in the reviewers' comments.

Li et al., present a study of aerosol characteristics conducted at rural site southwest of Houston during the intensive operation periods (IOP) of the Tracking Aerosol Convection Interactions ExpeRiment. They report significant differences in aerosol properties among marine, urban, and regional air masses, and identified six OA factors using PMF analysis. The results emphasize the roles of both atmospheric chemistry and meteorological conditions in this coastal-rural site. Overall, the manuscript is well-written and fits within the scope of the journal. However, the following comments need to be addressed before publication.

**1.** Line 185: Although the CWT map of ammonium is consistent with sulfate, the sources of ammonium could not from marine emissions. The authors should provide further discussion or references regarding potential terrestrial or anthropogenic sources of ammonium. Additionally, the organic nitrate mentioned also needs stronger supporting evidence.

We thank the reviewer for the suggestion. We agree that marine sources are unlikely to be a major contributor to ammonium. In the revised manuscript, we now clarify that the spatial similarity between ammonium and sulfate is most likely due to the formation of ammonium sulfate or bisulfate from terrestrial/anthropogenic sources, such as agricultural, industrial, and shipping emissions. Additional references have been included to support this interpretation.

We have revised this paragraph as suggested, on page 8, lines 208 to 211. It now reads as follows:

> Ammonium exhibits a similar spatial distribution to sulfate. However, because marine emissions are unlikely a major source of ammonia, the similarity likely reflects the formation of ammonium sulfate or bisulfate through atmospheric neutralization processes involving anthropogenic sulfate and terrestrial ammonia (Schiferl et al., 2014; Weber et al., 2016).

Due to the limited resolution of the ACSM, we cannot differentiate organic and inorganic nitrates. We have toned down the related discussion from 'nitrate is dominated by organic nitrate" to "nitrate may contain organic nitrate" in the revised text on page 8 line 213:

> The similar CWT and wind-rose patterns for nitrate and organics suggest that nitrate may contain organic nitrate (Fig. 4).

**2.** Section 3.2: The time series of all six PMF-resolved OA factors display high similarity, particularly during urban air mass periods. This raises questions about the robustness of the factor separation. For example, how to explain the co-variation of primary and secondary factors during the air mass from urban areas. In addition, how is the six-solution result determined? Providing diagnostics for this six-solution is beneficial for reader to understand.

We appreciate the reviewer raising this important issue. The co-variation among OA factors during urban air mass periods is likely due to the shared influence of strong urban emissions and meteorological transport.

However, despite some similarities in their time series, the mass spectral profiles and correlations with external tracers (e.g., $f_{91}$, $f_{82}$, $f_{44}$, $NO_3$, $SO_4$) remain distinct, supporting the physical interpretation of each factor. The HOA profile is dominated by fragments of aliphatic hydrocarbons. 91FAC has the highest $f_{91}$ and highly correlated with $NO_3$ ($R^2$=0.58). The isoprene-SOA exhibits the highest $f_{82}$ and correlates with $SO_4$ ($R^2$=0.36). The mass spectrum of OOA1 is highly correlated with that of organic aerosol strongly influenced by shipping emissions (Schulze et al., 2018). OOA2 and OOA3 show the highest O:C ratios among all 6 factors. The separation of OOA2 and OOA3 is supported by different O:C ratios and temporal variations of mass concentration (Fig. 5). From the wind-rose plot, both HOA and OOA1 concentration are elevated when the wind is from direction of the Houston Ship Channel, indicating a shared source. For 91FAC and isoprene-SOA (Fig. S17A), the hotspots in the direction of the Sam Houston National Forest suggest that they are SOA formed from biogenic emissions of the forest.

We applied the "rolling PMF" strategy (Canonaco et al., 2021) in this study. To guide the constrained rolling PMF analysis, we first conducted an unconstrained PMF analysis to explore the variability in potential factor profiles and identify suitable candidates for constraints. Multiple solutions with varying numbers of factors were tested, and repeated runs with random seeds were performed to evaluate solution stability. The results were clustered using the k-means methods, and silhouette analysis was used to assess the consistency of profiles within each solution. Based on this evaluation, three factors were selected as reference profiles: HOA, 91FAC, and isoprene-SOA. These factors were consistently observed across solutions and were chemically interpretable. Using these three factors as constraints, we then performed the rolling PMF analysis. For each rolling window, a random a-value between 0.1 and 0.6 was used to allow flexibility in the factor profiles. We evaluated three different rolling solutions: (1) a 4-factor solution with constrained HOA, 91FAC, and isoprene-SOA factors and one unconstrained OOA factor, (2) a 5-factor solution with the same constrained factors and two unconstrained OOA factors, and (3) a 6-factor solution with the same constrained factors and three unconstrained OOA factors. These discussions have been added to section 2.2 of the main manuscript text, on page 4, lines 120 to 130.

We consider the 6-factor solution optimal based on the mass spectral profiles and the correlations of the components with time series for tracer species. First, the mass spectral profiles of HOA are different in 3 solutions. The HOA factor in 6-factor solution exhibits a low O:C ratio (0.14, Fig. 5), consistent with previous studies (Docherty et al., 2011; Mohr et al., 2012). In contrast, the HOA factors in the 4- and 5-factor solutions have higher O:C ratio of 0.61 and 0.64 (Fig. S12A and Fig. S13A), respectively. Second, compared to 4-factor solutions, the 6-factor solution provides a more refined separation of OOA factors, yielding three distinct factors with different O:C ratios. Specifically, it resolves one factor associated with shipping emissions (OOA1) and two more oxidized secondary organic aerosol factors (OOA2 and OOA3). In contrast, the OOA factor in the 4-factor solution does not exhibit the characteristic signal as OOA1 in the 6-factor solution. Third, in the 6-factor solution, the correlation coefficient $R^2$ between isoprene-SOA and sulfate is 0.36, higher than those in the 4- and 5-factor solutions (0.25 and 0.27, respectively). Therefore, we consider the 6-factor solution to be the most physically meaningful and interpretable result for characterizing the organic aerosol sources in this study. Detailed information on the PMF resolution procedures and solutions comparison is presented in Section 1 of the Supplementary Information.

**3.** Line 215-225 paragraph: The attribution of 91FAC to monoterpene oxidation is reasonable based on spectral comparisons, but a few concerns remain: (1) the O/C ratio is higher in previous study using HR-ToF-AMS observation; (2) the name of 91FAC may cause confusion and a more descriptive term may improve clarity.

The slightly lower O/C ratio in our study, compared to values reported in HR-ToF-AMS studies, may arise from the different oxidation levels of OA. Different aging stages and oxidant levels at the rural site can

influence the O/C of the 91FAC. Perhaps the 91FAC here is fresher than previous studies. $NO_3$-initiated monoterpene SOA, particularly that formed under nighttime conditions or low OH exposure, can have low $f_{44}$ and still be secondary. Previous chamber studies also show a lower O/C ratio of the monoterpene oxidation product measured by HR-ToF-AMS (Boyd et al., 2015; He et al., 2021). The $f_{44}$ is ~2% in Boyd et al. (2015) and ~1% in He et al. (2021), comparable to ~1.4% in our study.

Regarding the naming, we agree that "91FAC" may be ambiguous and we have updated the name to "Monoterpene-derived SOA (MT-SOA)" in the revised manuscript to better reflect its source and chemical identity. We retained the original factor names in the response to reviewers to ensure consistency with the terminology used in the reviewers' comments.

**4.** Section 3.2.2: OOA1 is characterized by strong signals from m/z 55 and 57, which are typically associated with primary OA. However, the authors classify it as a secondary factor and link it to ship emissions. This conclusion needs further justification, such as its high O/C ratio and diurnal variation consistent with OOA2 and OOA3.

The mass spectrum of OOA1, characterized with strong signals at $m/z$ 55 and 57, agrees well with that of organics during periods of heavy shipping emissions reported by Schulze et al. (2018), with an $R^2$ value of 0.91 (Fig. S20). This suggests that OOA1 is likely associated with primary shipping emissions. However, this factor exhibits an O:C ratio of 0.60, which is higher than that typically observed for POA (Aiken et al., 2008; Hennigan et al., 2011). Collectively, this factor likely represents aged shipping emissions while still preserving some primary characteristics. Thus, to avoid the confusion of classifying it as either POA or SOA, we have renamed OOA1 as "shipping-OOA". We retained the original factor names in the response to reviewers to ensure consistency with the terminology used in the reviewers' comments.

**5.** OOA3: The decomposition of OOA3 by air masses is novel and provides useful insights. However, the explanation for the noon peak being due to longer land residence is somewhat inconsistent. Based on wind direction, southerly winds from the sea dominated during the daytime which contradicts the interpretation of increased urban influence. If the different diurnal variations between different air masses, it is generally not combined into one factor during the decomposition of PMF. In addition, the diurnal pattern of OOA3 within individual air mass is nearly flat, however the box model results show daytime production. Why is this happening?

We thank the reviewer for this insightful observation. The apparent inconsistency is due to the alternating dominance of land and sea breeze over different timescales. Although southerly winds prevail during midday, the backward trajectories reveal that air masses observed at this time have spent more hours over land prior to arrival (Fig. 8B). This increases their exposure to urban sources and explains higher OOA3 concentrations (Fig. S25B). To better explain this point, we included the calculated 24-hour backward trajectories based on the diurnal variation of wind direction and wind speed (Fig. R1). For example, at 13:00, when the wind originated from the south (i.e., the direction of the Gulf of Mexico), the air mass spent approximately 19 hours over the land. In contrast, at 5:00, although the wind shifted to the north (i.e., the urban direction), the air mass remained over land for about 12 hours. Therefore, even when the wind direction is from the south (i.e., from the gulf), the air mass spends more time over land compared to when the wind is from the north. In other words, wind direction alone does not represent the time the air mass spends over land.

We found that the diurnal variation of OOA3 remains relatively flat within both marine and urban air masses. This consistent temporal behavior supports the identification of OOA3 as a single factor by PMF.

As for the flat OOA3 trend within individual air masses, our box model shows that the magnitude of the overall chemical production/loss rate is low, below 0.05 µg m$^{-3}$ h$^{-1}$ at all times. In addition, the chemical production/loss is also partially offset by deposition and vertical mixing processes, further reducing the net change rate. This suggests that chemical production plays a minor role in the diurnal variation of OOA3, which is primarily due to shifting air mass types.

[Figure]

**Figure R1. The backward trajectories derived from the diurnal variations of wind direction and speed averaged over days with air mass changes during IOP (i.e. Fig. S25A). The red text on the left corner of each subplot is the time over the land of each trajectory. ANC site is marked by the cyan star.**

**Response to RC2:**

We thank the reviewers for their thoughtful and constructive comments. Please find below the response to each comment or question, including notations of improvements to the manuscript. Reviewer comments are in blue fonts and responses are in black. Changes to the manuscript are indented. We renamed the PMF factor "OOA1" to "shipping-OOA," to avoid the confusion of classifying it as either POA or SOA. Consequently, "OOA2" and "OOA3" were renamed as "OOA1" and "OOA2" in the revised main manuscript and supplementary information. We retained the original factor names in the response to reviewers to ensure consistency with the terminology used in the reviewers' comments.

Li et al. investigated aerosol properties in the southwest of Houston, a region that can be categorized as a coastal-rural site. The authors differentiated air masses using back-trajectory analysis and integrated this with OA PMF analysis to thoroughly examine the contributing factors to the observed amount and chemical characteristics of OA in the Houston area. Urban air masses were identified as the dominant driver of elevated aerosol levels, while marine air masses introduced ship emission plumes that led to enhanced OOA1. Notably, the authors highlight that OOA3, which is less likely to be locally influenced and is primarily composed of aged OA, is attributed to changes in air mass. This is demonstrated through back-trajectory and 1-D box model analysis—providing a different perspective from previous studies that pointed to photochemical formation or mixing from aloft. By utilizing multiple analytical approaches, the authors logically support their key findings, and the manuscript is overall well written. Still, I have a few comments that should be addressed before publication.

**Scientific Comments**

1. Line 166: Is the size at which the mode appears consistent with findings from previous studies?

Yes, the observed modal diameters in marine (60 and 150 nm) and urban (65 nm) air masses are consistent with previous observations in Houston (e.g., Levy et al., 2013; Schwarz et al., 2008). We have added this comparison in the text with citations, on page 7, lines 187 to 188. It reads now:

> The observed modal diameters in marine (60 and 150 nm) and urban (65 nm) air masses are consistent with previous observations in Houston (Levy et al., 2013; Schwarz et al., 2008).

2. Did the observed linear correlation align with the results from the CWT analysis? I think a statistical evaluation should come first before moving into trajectory-based analysis.

Yes, the observed linear correlation aligns with the results from the CWT analysis. The correlation coefficient $R^2$ is 0.54 between nitrate and organics, and 0.83 between sulfate and ammonium (Table S2). We have updated the manuscript to first present correlation analyses then proceed to the trajectory-based CWT to support spatial interpretation, on page 8, lines 203 to 205. It reads now:

> To investigate the origins of different aerosol components, we examined the correlation between each chemical component, conducted the CWT analysis, and examined the dependence of component concentrations on local wind speed and direction. Strong correlations were observed between nitrate and organics ($R^2 = 0.54$) and between sulfate and ammonium ($R^2 = 0.83$) (Table S2).

3. Line 214: Even with a strong tracer ion signal ($m/z$ 91), 91FAC appears to have a substantially low $f_{44}$, which makes it difficult to attribute this factor to SOA. Could you comment on this observation?

Thank you for this observation. While $f_{44}$ is typically used as a proxy for oxidation, certain $NO_3$-initiated SOA, particularly that formed under nighttime conditions or low OH exposure, can have low $f_{44}$ and still

be secondary. This has been observed in chamber studies (e.g., Boyd et al., 2015; He et al., 2021). The $f_{44}$ is ~2% in Boyd et al. (2015) and ~1% in He et al. (2021), comparable to ~1.4% in our study. Furthermore, despite the $f_{44}$ is low, 91FAC exhibits a high $f_{43}$ signal, another recognized tracer for SOA (Guo et al., 2020; Ng et al., 2011a). Combined with the strong correlation with $NO_3$ and the high $f_{91}$ signal, these features support the interpretation of 91FAC as a secondary OA factor.

4. Are there any potential isoprene sources besides the National Forest near the field site?

The dominant land cover in the immediate vicinity of the ANC site is grassland. Global estimates suggest that grasses and herbaceous plants emit much less isoprene than trees, contributing less than 4% of the total annual global isoprene emissions (Bai et al., 2006). Isoprene-SOA shows elevated mass concentration when the wind is from the northeast, the direction of Sam Houston National Forest (Fig. S17A). Therefore, while isoprene is emitted from grasslands nearby, forest emissions are likely the dominant isoprene source for the isoprene-SOA observed at the site. This is now noted in Section 3.2.1, on page 13, lines 262 to 266. Now it reads:

> The dominant land cover in the immediate vicinity of the ANC site is grassland. Global estimates suggest that grasses and herbaceous plants emit much less isoprene than trees, contributing less than 4% of the total annual global isoprene emissions (Bai et al., 2006). Isoprene-SOA shows elevated mass concentration when the wind is from the northeast, the direction of Sam Houston National Forest (Fig. S17A). Therefore, while isoprene is emitted from grasslands nearby, forest emissions are likely the dominant isoprene source for the isoprene-SOA observed at the site.

5. Since sulfate is often associated with ship emissions, do you think that maritime air masses with high $SO_4$ could also contribute to OOA1, not just isoprene OA formation? Given that OOA1 seems to be heavily influenced by coastal ship emissions, I'm curious about your interpretation here.

This is an insightful point. There is no evidence suggesting that high sulfate concentrations contribute to OOA1. We didn't observe $SO_4$ elevation during the OOA1 pollution event on September 19[th] (Fig. R1). In addition, the correlation between $SO_4$ and OOA1 is weak ($R^2 = 0.17$).

[Figure]

**Figure R1. An OOA1 pollution event from local time 4:00 to 8:00 on September 19, 2022. Time-series of mass concentration of (A)** OA factors **(B)** NR-PM$_1$ components.

6. You noted that the high $f_{55}/f_{57}$ signal of OOA in this study is likely linked to ship emissions rather than cooking. Would there be any common chemical species between the two sources? I think additional evidence would help support the classification of this factor as an OOA rather than a primary-like OA.

We appreciate this observation. The mass spectrum of OOA1, characterized with strong signals at $m/z$ 55 and 57, agrees well with that of organics during periods of heavy shipping emissions reported by Schulze et al. (2018), with an $R^2$ value of 0.91 (Fig. S20). This suggests that OOA1 is likely associated with primary shipping emissions. However, this factor exhibits an O:C ratio of 0.60, which is higher than that typically observed for POA (Aiken et al., 2008; Hennigan et al., 2011). Collectively, this factor likely represents aged shipping emissions while still preserving some primary characteristics. Thus, to avoid the confusion of classifying it as either POA or SOA, we have renamed OOA1 as "shipping-OOA". We retained the original factor names in the response to reviewers to ensure consistency with the terminology used in the reviewers' comments.

7. Line 237: Are you suggesting that these aerosols originate from different sources but become mixed near the site, subsequently forming isoprene OA?

Yes, that is correct. Our interpretation is that sulfate, primarily from the northeast (Houston Ship Channel), and isoprene, emitted from the north (Sam Houston National Forest), are transported via different pathways and become mixed downwind. In this mixed environment, the reactive uptake of IEPOX onto acidic sulfate particles facilitates the isoprene-SOA formation.

The production of isoprene-SOA depends on the availability of both isoprene and sulfate. In forested areas where isoprene emissions are spatially homogeneous and relatively constant, a strong correlation between isoprene-SOA and sulfate is often observed. However, at the ANC site, isoprene concentrations are more variable due to changing meteorological conditions, including wind direction and air mass transport. This spatial and temporal variability in precursor emissions and mixing conditions leads to day to day fluctuations in the timing and location of isoprene-SOA formation. As a result, the $R^2$ between isoprene-SOA and sulfate is relatively lower than previous studies observed in the Southeast U.S.

8. How did the size distribution vary during each OOA event, or when one OOA component was dominant over others?

Throughout the campaign, we identified one OOA1 event (Fig. R2), three OOA2 events (Fig. R3), and three OOA3 events (Fig. R4). Events were defined based on the following criteria: (1) the mass fraction of the specific OOA factor is the highest among all OA factors and above 30%, (2) the mass concentration of the specific OOA factor exceeded 2 µg m$^{-3}$, and (3) the elevated concentration persisted for at least three hours to exclude short-lived fluctuations.

During the OOA1 event, the particle volume size distribution exhibited a unimodal structure with a dominant mode around 250 nm. During the OOA2 events, two distinct types of size distributions were observed. The first type featured a mode around 150 nm (Fig. R3A), while the second type showed a more pronounced mode centered around 300 nm (Fig. R3D and R3G). Similarly, OOA3 events also displayed two characteristic patterns. The first type had a single mode near 250 nm (Fig. R4A and R4D). The second type exhibited a bimodal volume distribution, with one mode around 50 nm and the other around 250 nm (Fig. R4G). One OOA3 event was associated with the emergence of a new particle size distribution mode in the Aitken range (Fig. R4G). However, the appearance of this new mode may not be solely attributed to elevated OOA3 concentrations, as sulfate levels also increased during the same period. Therefore, it is possible that the observed particle mode resulted from particle growth influenced by the combined effects of sulfate and OOA3.

Overall, aerosol mass concentration shows an accumulation mode at ~ 250 nm during the vast majority of the OOA events. There is no significant change in aerosol size distribution associated with different OOA components.

[Figure]

**Figure R2. The OOA1 dominant event from local time 4:00 to 8:00 on September 19, 2022.** Time series of **(A)** particle volume size distribution, **(B)** mass concentrations of PMF factors and **(C)** mass concentration of NR-PM$_1$.

[Figure]

**Figure R3. Three OOA2 dominant events from local time 16:00 on August 11 2022 to 8:00 on August 12 2022, from local time 8:00 to 20:00 on August 25 2022, and from local time 18:00 on September 14 2022 to 16:00 on September 15 2022.** Time series of **(A, D, G)** particle volume size distribution, **(B, E, H)** mass concentrations of PMF factors and **(C, F, I)** mass concentration of NR-PM$_1$.

[Figure]

**Figure R4. Three OOA3 dominant events from local time 18:00 on June 20 2022 to 14:00 on June 21 2022, from local time 8:00 on June 25 2022 to 16:30 on June 26 2022, and from local time 11:00 to 19:00 on September 08 2022.** Time series of **(A, D, G)** particle volume size distribution, **(B, E, H)** mass concentrations of PMF factors and **(C, F, I)** mass concentrations of NR-PM₁.

9. Figure 7D: Since you conducted a CWT analysis, didn't that show an OOA1 hotspot over Freeport?

Thank you for this thoughtful question. The CWT analysis shows an OOA1 hotspot over Freeport in the marine air mass (Fig. R5). This is now added in Section 3.2.2, on page 16, lines 301 to 302. Now it reads:

> In addition, the CWT analysis shows shipping-OOA hotspots over Freeport for the marine air mass (Fig. S21).

[Figure]

**Figure R5. CWT analysis of OOA1 in marine air masses. The red square marks Freeport.**

10. The distinction between OOA2 and OOA3 would be more convincing if you could provide additional supporting evidence.

11. I understand the intent of this analysis is to support the idea that OOA3 originates as a background SOA. If that's the case, does the analysis yield a different interpretation for OOA1 and OOA2 if a similar approach is applied?

12. **Line 442:** What about OOA2? I think contrasting air mass trajectory characteristics could further support distinguishing OOA2 from OOA3.

Reply for questions 10, 11, and 12:

We thank the reviewer for these thoughtful questions and suggestions regarding the distinction between OOA2 and OOA3. The CWT analysis for urban air mass shows OOA2 hotspots over downtown Houston and the Houston Ship Channel, suggesting the influence from local urban emissions. In contrast, OOA3 displays a more spatially uniform distribution, indicating a more regional source (Fig. R6).

In addition, the diurnal patterns of these two factors under urban air mass conditions further illustrate their difference (Fig. R7). OOA3 exhibits relatively flat concentrations throughout the day, while OOA2 shows elevated levels during the nighttime. These differences in spatial distribution and temporal behavior support the interpretation that OOA2 is more locally influenced than OOA3. Moreover, the different O:C ratio and temporal variations of mass concentrations also indicate the distinction between OOA2 and OOA3 (Fig. 5).

Regarding Question 11, we presume the reviewer is referring to whether the 1-D box model approach could be applied to OOA1 and OOA2 as it was for OOA3. One of the assumptions underlying the box model is that the effect of horizontal advection is negligible. This assumption is not valid for OOA1 and OOA2, because OOA1 factor is strongly affected by local shipping emissions and OOA2 exhibits hotspots over downtown Houston and the Houston Ship Channel. Given the strong spatial variations and expected influences from horizontal advection, the same modeling framework cannot be applied to OOA1 and OOA2.

We have updated the content about OOA2, on page 16, lines 316 to 319. It now reads as follows:

> The CWT analysis for urban air mass shows OOA1 hotspots over downtown Houston and the Houston Ship Channel, suggesting the influence from local urban emissions. In contrast, OOA2 displays a more spatially uniform distribution, indicating a more regional source (Fig. S22). This interpretation is further supported by the variations of OOA1 and OOA2 mass concentrations with wind direction (Fig. S17).

[Figure]

**Figure R6. CWT analysis in the urban air mass of (A) OOA2 and (B) OOA3. The red triangle marks the sampling site.**

[Figure]

**Figure R7. Diurnal variations of OA factors mass concentrations during the IOP in marine and urban air masses.**

**Technical Comments**

**Methods**

1. Have you deployed a cyclone for the size cut of sampling aerosol? Was ACSM equipped with a standard vaporizer or a capture vaporizer? Please clarify.

We deployed a cyclone with 2.5 µm cut off upstream of the ACSM. The ACSM was equipped with a standard vaporizer. We added it to the main manuscript section 2.1, on page 4, lines 106 to 107. It reads now:

> The chemical composition of NR-PM$_1$ was measured using a Time of Flight - Aerosol Chemical Speciation Monitor (ToF-ACSM, *Aerodyne Research*) with a standard vaporizer (Fröhlich et al., 2013; Watson, 2017). A cyclone with a cut size of 2.5 µm was installed upstream of the ACSM inlet.

2. Line 105: Cite Fröhlich et al., 2013.

Thanks for this suggestion. We added the citation, on page 4, line 106. It reads now:

> The chemical composition of NR-PM$_1$ was measured using a Time of Flight - Aerosol Chemical Speciation Monitor (ToF-ACSM, *Aerodyne Research*) with a standard vaporizer (Fröhlich et al., 2013; Watson, 2017).

3. Line 111: Could you provide more details on the PMF analysis? Specifically, how was the solution selected, and how did the analysis results vary depending on the number of factors?

We applied the "rolling PMF" strategy (Canonaco et al., 2021) in this study. To guide the constrained rolling PMF analysis, we first conducted an unconstrained PMF analysis to explore the variability in potential factor profiles and identify suitable candidates for constraints. Multiple solutions with varying numbers of factors were tested, and repeated runs with random seeds were performed to evaluate solution stability. The results were clustered using the k-means methods, and silhouette analysis was used to assess the consistency of profiles within each solution. Based on this evaluation, three factors were selected as reference profiles: HOA, 91FAC, and isoprene-SOA. These factors were consistently observed across solutions and were chemically interpretable. Using these three factors as constraints, we then performed the rolling PMF analysis. For each rolling window, a random a-value between 0.1 and 0.6 was used to allow flexibility in the factor profiles. We evaluated three different rolling solutions: (1) a 4-factor solution with constrained HOA, 91FAC, and isoprene-SOA factors and one unconstrained OOA factor, (2) a 5-factor solution with the same constrained factors and two unconstrained OOA factors, and (3) a 6-factor solution with the same constrained factors and three unconstrained OOA factors. These discussions have been added to section 2.2 of the main manuscript text, on page 4, lines 120 to 130.

We consider the 6-factor solution optimal based on the mass spectral profiles and the correlations of the components with time series for tracer species. First, the mass spectral profiles of HOA are different in 3 solutions. The HOA factor in 6-factor solution exhibits a low O:C ratio (0.14, Fig. 5), consistent with previous studies (Docherty et al., 2011; Mohr et al., 2012). In contrast, the HOA factors in the 4- and 5-factor solutions have higher O:C ratio of 0.61 and 0.64 (Fig. S12A and Fig. S13A), respectively. Second, compared to 4-factor solutions, the 6-factor solution provides a more refined separation of OOA factors, yielding three distinct factors with different O:C ratios. Specifically, it resolves one factor associated with shipping emissions (OOA1) and two more oxidized secondary organic aerosol factors (OOA2 and OOA3). In contrast, the OOA factor in the 4-factor solution does not exhibit the characteristic signal as OOA1 in the 6-factor solution. Third, in the 6-factor solution, the correlation coefficient $R^2$ between isoprene-SOA and sulfate is 0.36, higher than those in the 4- and 5-factor solutions (0.25 and 0.27, respectively). Therefore, we consider the 6-factor solution to be the most physically meaningful and interpretable result for characterizing the organic aerosol sources in this study. Detailed information on the PMF resolution procedures and solutions comparison is presented in Section 1 of the Supplementary Information.

**Figures & Organization**

1. Fig. 9 appears before Fig. 8 in the main text. Please reorganize the figures to appear in sequential order.

2. Since Fig. 9 is based on box modeling and Fig. 8 uses backtrajectory analysis, I suggest not introducing Fig. 9 before Fig. 8. Instead, mention that Fig. 9 supports the analysis presented in Fig. 8.

3. I think this explanatory section should come before the discussion of Figs. 8 and 9 to maintain the current flow. (I'd prefer it in the Method section, but as long as it appears before those figure discussions, I'll leave that to the authors.)

Reply for questions 1, 2, and 3:

We appreciate these thoughtful suggestions. We have modified Fig. 8. Figure 9 (box model) and its assumptions are introduced following the presentation of Fig. 8, as the support of Fig. 8. These changes improve the logical flow. Fig. 8 shows now:

[Figure]

**Figure 8. (A) Diurnal variations of observed OOA2 mass concentrations in marine, urban, and unseparated air mass; (B) Diurnal variations of observed OOA2 mass concentrations and the time spent by the air mass over land during the 24 hours before arriving at the site; (C) Diurnal variations of observed OOA2 mass concentration and the percentage of the urban air mass. The shaded areas in A and B represent ± one standard deviation.**

The relevant texts are on page 16, lines 330 to 332. It now reads:

> For each individual air mass type (i.e., marine or urban), the OOA2 mass concentration largely remains constant throughout the day (Fig. 8A), in contrast to the diurnal variation of OOA2 in all air masses (referred to as "unseparated air masses" thereafter).

With the modification of Fig. 8 and relevant text, Fig. 9 is now introduced following Fig. 8, and the 1-D box model is detailed before Fig. 9, which improves the flow.

4. Line 247: Please also note that OOA separation typically has greater uncertainties compared to POAs (Zhang et al., 2011; Ng et al., 2011; Hass-Mitchell et al., 2024).

We added a statement acknowledging this uncertainty and cited Zhang et al. (2011), Ng et al. (2011), and Hass-Mitchell et al. (2024), on page 16, lines 307 to 310. It reads now:

> Two oxygenated OA factors with different O:C ratios (1.05 and 1.36 for OOA1 and OOA2, respectively) were identified, collectively accounting for 53% of OA mass concentration (Fig. 6A). As the OA mass spectra become increasingly similar through the aging process, separating OOA factors and identifying their precursors and sources is more challenging compared to other OA factors (Hass-Mitchell et al., 2024; Ng et al., 2011b; Zhang et al., 2011).

5. Line 345: Are you referring to Fig. 9C and 9F? Please clarify which figure panels are being discussed.

We revised the text to clearly reference "Fig. 9C and Fig. 9F", on page 18, line 395. It reads now:

[revised manuscript text omitted]